# Electronic media use and symptoms of depression among adolescents in Norway

**Annette Løvheim Kleppang**[1]*, **Anne Mari Steigen**[2], **Li Ma**[1,3], **Hanne Søberg Finbråten**[2], **Curt Hagquist**[1,4]

**1** Department of Public Health and Sport Sciences, Faculty of Social and Health Sciences, Inland Norway University of Applied Sciences, Hamar, Norway, **2** Department of Health and Nursing Sciences, Faculty of Social and Health Sciences, Inland Norway University of Applied Sciences, Hamar, Norway, **3** Centre for Research on Child and Adolescents Mental Health, Karlstad University, Karlstad, Sweden, **4** Department of Education and Special Education, University of Gothenburg, Gothenburg, Sweden

* annette.kleppang@inn.no

## Abstract

### Background

The purpose of this study was to examine the association between electronic media use, including use of social media and gaming, and symptoms of depression, and whether gender or having friends moderated these associations.

### Methods

This study was based on self-reported cross-sectional data from the Ungdata survey, conducted in 2018 by the Norwegian Social Research (NOVA) Institute in cooperation with seven regional drug and alcohol competence centres. The target group comprised 12,353 15–16 years old adolescents. Binominal logistic regression was used to analyse the association between electronic media use and symptoms of depression.

### Results

The odds of having symptoms of depression were higher for those who used social media more than 3 hours per day (OR: 1.60, 95% CI: 1.43–1.80), compared to those who used social media 3 hours or less per day. Additionally, the odds of having symptoms of depression was higher for those who used more than 3 hours on gaming per day (OR: 1.57, 95% CI: 1.36–1.80), compared to those who used 3 hours and less on gaming per day after adjustment for potential confounders. There were no interaction effects between social media and gaming use with symptoms of depression. Neither were the associations between social media use and gaming with symptoms of depression moderated by gender or having friends.

### Conclusions

The odds of having symptoms of depression were significantly higher for adolescents with a more frequent use of electronic media.

**Data Availability Statement:** The data and materials from the Ungdata Surveys are stored in a national database administered by NOVA. The data are available for research purposes upon

application. For request of the data, please contact ungdata@oslomet.no. Further information about the study and the questionnaires can be found on the web page https://www.nsd.no/nsddata/serier/ungdata_eng.html.

**Funding:** The authors received no specific funding for this work.

**Competing interests:** The authors have declared that no competing interests exist.

## Background

Mental health problems are a growing concern worldwide [1], and studies on trends among adolescents have reported an increase across countries [2, 3]. The number of adolescents reporting symptoms of depression has also increased in Norway, with a higher number of girls reporting symptoms of depression than boys [4]. Depression represents a major public health challenge [5], and 19% of adolescents (aged 12–19) in Norway report that they are much affected by depressive symptoms [4]. The increasing prevalence of depressive symptoms signifies a need to identify modifiable risk factors.

The concerns about the heavy influence screen use have on children and adolescents are growing, and one of the concerns is related to mental health problems [6]. Adolescents nowadays spend more time on gaming and social media than previous generations [7]. In Norway, a recent survey reported that 66% of Norwegian adolescents aged 15–16 years spent more than 3 hours a day on screen-based activities (Bakken 2019). While girls are mostly active on social media, boys are mostly engaged in gaming [8].

Systematic reviews report that among adolescents, a higher level of electronic media use is associated with depressive symptoms and psychological distress [9, 10]. Additionally, several studies report associations between increased time spent on social media and higher levels of depression [11–13], while other studies question the association between time spent on social media and depressive symptoms [14–16]. Positive outcomes of social media use are also reported [17]. On the other hand, a longitudinal study among US adolescents found no association between increased time on social media and increased depression and anxiety [14].

Some studies on video gaming have reported no association between video gaming and symptoms of depression [11, 18, 19]. In contrast, a study among Canadian adolescents showed that both video gaming and computer use were associated with depressive symptoms [20]. In the most recent decades, both depressive symptoms and time spent on electronic media have been increasing simultaneously [4]. Therefore, it is of great importance to study the association between these trends. Moreover, different types of screen use behaviour may have different impacts on depressive symptoms [21].

Social media use, video gaming and computer use are more likely to be associated with depressive symptoms compared to television viewing [22]. Zink et al. [22], indicated in a recent systematic review of moderating variables, that screen-type influences the strength of the association between screen based sedentary behaviour and symptoms of depression and anxiety in adolescence. Some previous studies have not found any modifying effect of gender on the association between social media use and depression [12, 23], whereas a recent longitudinal study did find gender being a possible moderator of the associations between video game use and depressive symptoms [24]. Another study which used objective measures of video gaming found positive associations with wellbeing [25]. Social support is suggested as one of the protective factors potentially influencing the relationship between social media use and depression [26].

A gap in previous research is the lack of information on potential moderators of the association between electronic media use and depressive symptoms. The aim of this study is therefore to examine the association between electronic media use and symptoms of depression, and to study whether gender and having friends moderate the association between electronic media use and symptoms of depression.

## Methods

This study was based on data from 2018 retrieved from the Ungdata surveys conducted by the Norwegian Social Research (NOVA) institute in cooperation with regional drug and alcohol

competence centres (KoRus). The Norwegian Directorate of Health, The Ministry of Children, Equality and Social Inclusion, and the Ministry of Justice and Public Security have financed the surveys. Ungdata is a repeated cross-sectional survey and have been conducted in many secondary schools all over the country since 2010. The Ungdata survey covers different aspects of young people's lives, such as media use, physical activity, health issues, relationships with friends and parents, local environment, school issues, depressive symptoms, and wellbeing. In this study, we included adolescents of grade 10 (aged 15–16 years) from secondary schools all over the country. The web-based questionnaire was administered anonymously at school during a school hour with a teacher present available for questions. The parents received written information in advance through the school learning portal. Also, adolescents completed a consent form before participation and were informed that participation was voluntary. Altogether 12,353 adolescents in grade 10 participated, and the overall response rate in the secondary schools was 85% [27]. The study was conducted in line with the Declaration of Helsinki, and the data were analysed by independent researchers who did not participate in the data collection. The Norwegian Centre for Research Data (NSD) approved all ethical aspects of the study.

Table 1 gives the Ungdata Survey 2018 questions, response alternatives and variable definitions included in this study.

## Measures

**Symptoms of depression.** Symptoms of depression were measured with six items. The items originate from the Depressive Mood Inventory [28] which was derived from the Hopkins Symptom Checklist—90 [28, 29]. The six items are used both as single items and to construct a scale that provides a total score for depressive symptoms [30, 31]. The adolescents were asked if during the past week they have been affected by any of the following: "Felt that everything is a struggle (item 1)", "had sleep problems (item 2)", "felt unhappy, sad or depressed (item 3)", "felt hopelessness about the future (item 4)", "felt stiff or tense (item 5)", worried too much about things (item 6)". The six questions have four response categories: "Not been affected at all (1)", "not been affected much (2)", "been affected quite a lot (3)" and "been affected a great deal (4)".

The Rasch Measurement Theory [32, 33] has been used to examine the psychometric properties of the depressive symptoms scale in Ungdata from 2017 [34, 35] and in the present study. The scale shows good reliability (Person Separation Index: 0.82). The items work relatively well at a general level, except for item 2, 'Sleeping problems', which clearly misfit. The DIF-analysis indicated evidence of DIF for some items using ANOVA based on standardized residual of each person to each item. While there are some weaknesses, as a whole the symptoms of depression scale work well.

In the Rasch analysis the non-linear raw scores were transformed to person estimates on a linear interval logit scale on which each person is allocated a location (logit) value. In the present study, these person estimates were used in the statistical analysis. Lower values on the scale indicate a lower degree of symptoms of depression. For the purpose of this study, the symptoms of depression scale was dichotomized based on the latent symptoms of depression variable generated by the Rasch analysis. A cut off point, reported as logit value, was set at the 80th percentile (0.522), implying two categories on the scale: a) symptoms of depression ($\geq$80th percentile) and b) no symptoms of depression (<80th percentile).

**Electronic media use.** Electronic media use was measured using the two variables *Social media use* and *Gaming*. Social media use was measured by asking: "Think about what you do a normal day, how much time do you spend on the following things: social media (Facebook, Instagram etc.)", and categorized as no time, < 30 minutes, 30 minutes– 1 hour, 1–2 hours,

**Table 1. Ungdata Survey 2018: Questions, response alternatives and variable definitions.**

| Questions | Response alternatives | Variable definitions |
|---|---|---|
| *Symptoms of depression* | | |
| During the past week, have you been affected by any of the following issues: | | |
| Felt that everything is a struggle (item 1) | Not been affected at all, not been affected much, been affected quite a lot and been affected a great deal. | Symptoms of depression $\geq$ 80th percentiles |
| Had sleep problems (item 2) | | |
| Felt unhappy, sad or depressed (item 3) | | |
| Felt hopelessness about the future (item 4) | | |
| Felt stiff or tense (item 5) | | |
| Worried too much about things (item 6) | | |
| *Social media use* | | |
| Think about what you do a normal day: How much time do you spend on the following things: | No time, < 30 minutes, 30 minutes—1 hour, 1–2 hours, 2–3 hours, < 3 hours | $\leq$ 3 hours per day, > 3 hours per day |
| Social media (facebook, Instagram etc.) | | |
| *Gaming on computer/TV, telephone/tablets* | | |
| Think about what you do a normal day: How much time do you spend on the following things: | No time, < 30 minutes, 30 minutes—1 hour, 1–2 hours, 2–3 hours, < 3 hours | Gaming,: $\leq$ 3 hours per day, > 3 hours per day |
| gaming on computer/TV? | | |
| *Friends* | | |
| Do you have at least one friend who you trust completely and who you can tell absolutely anything? | Yes definitely, yes I think so, I don´t think so and I have nobody I would call close online friends at the moment | Yes, no |
| *Gender* | | |
| Are you a boy or a girl? | Boy, girl | |
| *Smoking* | | |
| Do you smoke? | I've never smoked, I used to smoke but I've stopped completely now, I smoke less than once a week, I smoke every week but not every day and I smoke every day. | No smoking, smoking |
| *Parents' higher education* | | |
| Did your father and mother go to university or to a university college? Select one answer for each parent. If you are not in touch with one or both of your parents, then skip the question about that parent. | Yes, no | Both parents, One of the parents, None of the parents |
| *Family economy* | | |
| Financially, has your family been well off, or badly off, over the past years? | We have been well off the whole time, we have generally been well off, we have neither been well off nor badly off, we have generally been badly off, we have been badly off the whole time | Good economy, neither bad nor good economy, bad economy |

2–3 hours and > 3 hours. Social media use was in the present study dichotomized as "more than 3 hours" and "3 hours or less". Gaming was measured by asking: "Think about what you do a normal day, how much time do you spend on the following things: gaming on telephone/tablets or gaming on computer/TV", and categorized as no time, < 30 minutes, 30 minutes– 1 hour, 1–2 hours, 2–3 hours and > 3 hours. Gaming was in the present study dichotomized as "more than 3 hours" and "3 hours or less".

**Friends.** Friends were measured by asking: "Do you have at least one friend who you trust completely and who you can tell absolutely anything?" This was categorized as Yes, definitely, Yes, I think so, I don´t think so, and I have nobody I would call close online friends at the moment. This was in the present study categorized as "Yes, I have friends" and "No, I do not have friends".

**Smoking.** Smoking was measured by asking: "do you smoke?" It was categorized as: I've never smoked, I used to smoke but I've stopped completely now, I smoke less than once a week, I smoke every week but not every day and I smoke every day. Smoking was dichotomized as"no smoking" and "smoking".

**Parents' level of education and family economy.** Parents' level of education was measured separately for each parent by asking the following: "Did your father/mother go to university or to a university college?" Those young people not in touch with one or both parents, were asked to miss the question out. This was categorized as yes or no. Parental educational status was stratified as "no university education", "one parent with university education" or "both parents with university education". Family economy was measured as follows: "financially, has your family been well off, or badly off, over the past years?" This was categorized as: we have been well off the whole time, we have generally been well off, we have neither been well off nor badly off, we have generally been badly off, we have been badly off the whole time. Family economy was stratified as "good", "neither bad nor good", and "bad economy".

## Analysis

Descriptive contingency table and logistic regression analyses were conducted using SPSS 24.0 for Windows. In the descriptive analyses, the study population was stratified according to symptoms of depression and gender. Baseline characteristics were presented as proportions with 95% confidence intervals (CI) in each stratum. No overlap of the CI was considered significant at the 5% level.

Binomial logistic regression analysis was performed to examine the association between electronic media use (social media and gaming) and symptoms of depression, adjusted for possible confounding variables. A p-value of $\leq 0.05$ was set as the level for statistical significance. Associations were presented as odds ratios (OR) with 95% CI, and with adjustments for friends, gender, smoking, parent's education, and family economy.

Interaction analyses were used to examine whether gender and having friends affected the strength of the relationship between electronic media use and symptoms of depression. Similarly, potential interaction between social media use and gaming was examined. All potential interaction effects were examined using the Likelihood Ratio test (LR test), contrasting models with and without interaction terms. The main effect model included social media use, gaming, friends, gender, smoking, parents' education and family economy as independent variables, and was tested against models including the following interactions; social media* gaming, social media *gender, social media *friends, gaming*gender, gaming*friends, and social media*gaming *gender. The incremental changes in log-likelihood between the main effect models and models including interactions were not significant, implying that the fit was not improved when applying interaction models. Therefore, only the main effect model is presented in the results section.

To check the robustness of our results, we tried dichotomizing the outcome variable regarding time spent on electronic media into more than two hours and two hours or less.

## Results

Table 2 shows baseline characteristics of the study population aged 15–16 years in 2018, according to depressive symptoms and gender.

Overall, a significantly higher proportion of the girls (28.2%) had symptoms of depression compared with the boys (10.3%). In gender subgroups, those with symptoms of depression reported significantly poorer family economy, more smoking, fewer friends and that their parents had lower education, compared with the rest of the study population.

Among adolescents who spent more than 3 hours per day on gaming and social media, a higher proportion reported symptoms of depression compared with adolescents using 3 hours and less per day. There were small gender differences. Of those who reported using social media more than 3 hours per day, a higher proportion of the girls (46.8% compared to 34.7%)

**Table 2. Youth data surveys: Baseline characteristics of adolescents, aged 15–16 years in 2018, according to symptoms of depression and gender.**

| | Total (N = 11836) | | Boys (N = 6133) | | | Girls (6169) | | |
|---|---|---|---|---|---|---|---|---|
| | No symptoms of depression | Symptoms of depression | Total | No symptoms of depression | Symptoms of depression | Total | No symptoms of depression | Symptoms of depression |
| | (N = 9528), n (%; 95% CI) | (N = 2308), n (%; 95% CI) | (N = 5777), n (%; 95% CI) | (N = 5181), n (%; 95% CI) | (N = 596), n (%; 95% CI) | (N = 6013), n (%; 95% CI) | (N = 4320), n (%; 95% CI) | (N = 1693), n (%; 95% CI) |
| **Electronic media use** | | | | | | | | |
| *Social media (N = 11633)* | | | | | | | | |
| > 3 hours | 2346 (25.0; 24.2–25.9) | 958 (42.4; 40.4–44.3) | 1027 (18.2; 17.2–19.2) | 862 (17.0; 15.9–18.0) | 165 (28.9; 25.2–32.7) | 2263 (38.1; 36.9–39.3) | 1480 (34.7; 33.2–36.1) | 783 (46.8; 44.4–49.2) |
| *Gaming (N = 11423)* | | | | | | | | |
| > 3 hours | 1812 (19.7; 18.9–20.5) | 507 (22.8; 21.1–24.6) | 1655 (29.8; 28.6–31.0) | 1419 (27.2; 26.2–29.7) | 236 (42.0; 37.9–46.1) | 654 (11.2; 10.4–12.0) | 389 (9.3; 8.4–10.2) | 265 (16.2; 14.4–18.0) |
| **Confounding variables** | | | | | | | | |
| **Friends (N = 11780)** | | | | | | | | |
| Yes | 8448 (89; 88.0–89.3) | 1880 (81.0; 79.8–83.0) | 5744 (89.8; 89.0–90.6) | 4704 (91.1; 90.4–91.9) | 455 (78.2; 74.8–81.5) | 5332 (89.0; 88.2–89.2) | 3921 (91.1; 90.2–91.9) | 1411 (83.6; 81.8–85.4) |
| *Smoking N = 11836* | | | | | | | | |
| No smoking | 8847 (92.9; 92.3–93.4) | 1967 (85.2; 83.7–86.6) | 5218 (90.3; 89.6–91.1) | 4748 (91.6; 90.9–92.4) | 470 (78.9; 75.3–82.0) | 5635 (94.0; 93.1–94.3) | 4073 (94.3; 93.6–95.0) | 1481 (87.5; 85.9–89.1) |
| Smoking | 681 (7.1; 6.6–7.7) | 341 (14.8; 13.3–16.2) | 559 (9.7; 8.9–10.4) | 433 (8.4; 7.6–9.1) | 126 (21.1; 17.0–24.4) | 459 (7.6; 7.0–8.3) | 247 (5.7; 5.0–6.4) | 212 (12.5; 11.0–14.1) |
| *Parents' higher education N = 10164* | | | | | | | | |
| Both parents | 4982 (60.7; 59.7–61.8) | 1059 (54.1; 51.9–56.3) | 2952 (59.3; 57.9–60.6) | 2675 (60.0; 58.5–61.4) | 277 (53.5; 49.2–57.8) | 3066 (59.5; 58.2–60.9) | 2294 (61.6; 60.0–63.1) | 772 (54.3; 51.7–56.8) |
| One of the parents | 1731 (21.1; 20.2–22.0) | 452 (23.1; 21.2–25.0) | 1040 (20.9; 19.8–22.0) | 928 (20.8; 19.6–22.0) | 112 (21.6; 18.1–25.2) | 1139 (22.1; 21.0–23.3) | 801 (21.5; 20.2–22.8) | 338 (23.8; 21.5–26.0) |
| Neither parents | 1494 (18.2; 17.4–19.0) | 446 (22.8; 20.9–24.7) | 988 (19.8; 18.7–21.0) | 859 (19.3; 18.1–20.4) | 129 (24.9; 21.2–28.6) | 944 (18.3; 17.3–19.4) | 631 (16.9; 15.7–18.1) | 313 (22.0; 19.8–24.2) |
| *Family economy N = 11672* | | | | | | | | |
| Good economy | 7653 (81.4; 80.6–82.2) | 1471 (64.9; 62.9–66.8) | 4600 (80.5; 70.5–81.6) | 4214 (82.2; 81.1–83.2) | 386 (66.1; 62.3–69.9) | 4495 (76.0; 74.9–77.0) | 3422 (80.5; 79.3–81.7) | 1073 (64.4; 62.2–66.7) |
| Neither bad nor good | 1405 (14.9; 14.2–15.7) | 547 (24.1; 22.4–25.9) | 869 (15.2; 14.3–16.1) | 734 (14.3; 13.4–15.3) | 135 (23.1; 19.7–26.5) | 1074 (18.1; 17.2–19.1) | 666 (15.7; 14.4–16.8) | 408 (24.5; 22.4–26.6) |
| Bad economy | 346 (3.7; 3.3–4.1) | 250 (11.0; 9.7–12.3) | 243 (4.3; 3.7–4.8) | 180 (3.5; 3.0–4.0) | 63 (10.8; 8.3–13.3) | 349 (5.9; 5.3–6.5) | 165 (3.9; 3.3–4.5) | 184 (11.1; 9.6–12.6) |

Symptoms of depression coded as $\geq$ 80th percentiles, no symptoms of depression coded as < 80th percentiles.

and a higher proportion of boys (28.9% compared to 17.0%) reported symptoms of depression. Of those who reported gaming more than 3 hours per day, a higher proportion of the girls (16.2% compared to 9.3%) and a higher proportion of boys (42.0% compared to 27.2%) reported symptoms of depression.

Table 3 present binominal logistic regression of symptoms of depression in relation to electronic media use and other factors.

Table 3 presents the odds ratios of depressive symptoms by social media (column 1), gaming (column 2) and social media use and gaming (column 3) adjusted for potentially confounding variables. In the multivariate model where gender, friends, smoking, parents' higher

**Table 3. Binary logistic regression analysis of symptoms of depression in relation to social media use and gaming, controlling for possible confounders factors (Ungdata 2018, adolescents aged 15–16 years).**

|  | Analysis 1, social media use | Analysis 2, gaming | Analysis 3, social media use and gaming |
|---|---|---|---|
| **Variables** | AOR (95% CI) | AOR (95% CI) | AOR (95% CI) |
| **Social media use** |  |  |  |
| 3 hours or less | 1 (ref) |  | 1 (ref) |
| More than 3 hours | 1.60 (1.43–1.80) |  | 1.51 (1.34–1.70) |
| **Gaming** |  |  |  |
| 3 hours or less |  | 1 (ref) | 1 (ref) |
| More than 3 hours |  | 1.57 (1.36–1.80) | 1.38 (1.19–1.59) |

Symptoms of depression coded as ≥ 80th percentiles, no symptoms of depression coded as < 80th percentiles. OR: odds ratio; 95% confidence interval. AOR: adjusted for gender, friends, smoking, parent's higher education and family economy.

education and family economy were adjusted, the odds of having symptoms of depression were 1.60 times higher for those who use social media more than 3 hours per day, compared to those who use social media 3 hours and less per day. Additionally, the odds of having symptoms of depression were higher for those who spent more than 3 hours on gaming per day (OR: 1.57, 95% CI: 1.36–1.80), compared to those who spent 3 hours and less on gaming per day. These odds were little changed when the two variables were entered simultaneously. There was no interaction effect with gender and friends according to the likelihood ratio test.

To check the robustness of our results, we tried dichotomizing the outcome variable regarding time spent on electronic media into more than two hours and two hours or less. The results showed that adolescents who spent more than two hours on social media every day had higher odds of depressive symptoms than those who spent two hours or less. Further, adolescents who spent more than two hours in gaming had higher odds of depressive symptoms compared to those who spent two hours or less on the same activity.

## Discussion

This study examined the association between electronic media use, including use of social media and gaming, and symptoms of depression among adolescents in Norway, and whether gender or having friends moderated these associations. Our results based on multivariate analyses showed that symptoms of depression was significantly associated with adolescents' frequent use of social media and gaming after adjustment for having friends, gender, smoking, parents higher education and family economy.

These results are consistent with many previous studies [9, 10]. However, some studies reported no association between gaming and depressive symptoms [11, 18, 19]. Some other studies found a protective effect of video gaming on depression [36]. Moreover, using objective measures of video gaming a recent cross-sectional study found a positive relationship between video gaming and affective well-being [25]. The inconsistent results may depend on the purpose of playing the videogame, the content and the type [22].

There were no interaction effects between use of social media and gaming. This means that the strength of the association between social media use and symptoms of depression was not stronger among frequent game players than among non-frequent gameplayers. Inversely, the association between gaming and symptoms of depression is as strong among frequent social media users as among non-frequent social media users. These patterns also held when the analyses were repeated separately for boys and girls. Moreover, the analyses did not show any

interactions between electronic media use and having a friend. According to Zink et al.[22], previous studies provide inconsistent results for moderating effect of gender, friendship, physical activity, social context, cultural characteristics, age, parental factors, and no moderating effect of socioeconomic status.

Another study showed that sleep mediated the association between screen time (eg, social messaging, web surfing, TV/movie watching, and gaming) and depressive symptoms among adolescents, but they did not examine interaction effects [37].

The discrepancy in the results may be explained by that the adolescents spend more time on electronic media use compared to previous generations [7]. Additionally, the discrepancy may reflect the increasing availability of screen based devices, and that type and features of online activities might be associated with mental health differently [38]. A longitudinal study among Norwegian adolescent showed a discrepancy in the results due to measurement methods, self-reported screen time was associated with mental distress while objectively measured sedentary behaviour was not [39].

To test the robustness of our results, we dichotomized the outcome variable into more than two hours and two hours or less. The estimated results show that those who spent more than two hours on social media or gaming every day have higher odds of depressive symptoms than those who spent two hours or less. The results are consistent with the results presented in this study, signaling that spending longer time on electronic media use is associated with higher odds of having symptoms of depression, all else equal.

The contribution of this study lies in at least two aspects. First, there is a lack of knowledge about the association between electronic media use and depressive symptoms among adolescents in the Nordic countries. Based on the most recent Norwegian data, this study will enrich our relevant knowledge of the situation in the Nordic area. Second, findings of this study may guide policy development regarding mental health issues of children and adolescents.

The cross-sectional study design does not allow conclusions about the direction of the association, i.e., whether excessive social media use caused depressive symptoms or if adolescents with previous mental health problems are overrepresented among excessive media users. Alternatively, the association may work in both directions.

## Strength and limitations

A major strength of this study is the large sample size in combination with a high response rate. The data used in this study were collected in 2018 and provide an up-to-date description of key aspects of young people's lives. Limitations of this study exist. First of all, this is a cross-sectional study, which precludes inferences about causal relationships. Depressive symptoms may not only work as an outcome but may also act as an exposure that is hampering electronic media use. Furthermore, the use of self-reported measures may have led to unidentified misclassifications or measurement errors. As far as we know, no evaluation has been performed in the Ungdata Survey regarding self-reporting of electronic media use. Furthermore, the associations found between electronic media use and depressive symptoms might be influenced by other factors that were not controlled for in the present study. For example, our data do not include information regarding the content of social media and gaming. We had no information about what adolescents were doing on social media or what types of games that they were playing. Thus, our results should be interpreted with caution.

## Conclusions

Adolescents with a more frequent use of electronic media was associated with significantly higher odds of having symptoms of depression. Our results did not vary by gender or having

friends or not. From a public health perspective, we hope these results can contribute to the discussion regarding the relationship between electronic media use and mental health. Future research should aim at including variables that consider the context and content of electronic media use–and whether and how these factors might influence the relationship between electronic media use and symptoms of depression.

## Author Contributions

**Conceptualization:** Annette Løvheim Kleppang, Li Ma, Curt Hagquist.

**Formal analysis:** Annette Løvheim Kleppang.

**Project administration:** Curt Hagquist.

**Writing – original draft:** Annette Løvheim Kleppang, Anne Mari Steigen.

**Writing – review & editing:** Anne Mari Steigen, Li Ma, Hanne Søberg Finbråten, Curt Hagquist.

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
