## [Decision Letter · Decision Letter 0]

20 Apr 2021

PONE-D-20-34067

Electronic media use and symptoms of depression among adolescents in Norway

PLOS ONE

Dear Dr. Kleppang,

Thank you for submitting your manuscript to PLOS ONE and apologies for the delay in responding. It has proven difficult to find reviewers but fortunately 2 reviewers have agreed to review your article. After careful consideration, we feel that it has merit but does not fully meet PLOS ONE’s publication criteria as it currently stands. Therefore, we invite you to submit a revised version of the manuscript that addresses the points raised during the review process.

The two reviewers still have some minor comments that you would need to address.

We look forward to receiving your revised manuscript.

Kind regards,

Therese van Amelsvoort

Academic Editor

PLOS ONE

Journal Requirements:

2a) If there are ethical or legal restrictions on sharing a de-identified data set, please explain them in detail (e.g., data contain potentially identifying or sensitive patient information) and who has imposed them (e.g., an ethics committee). Please also provide contact information for a data access committee, ethics committee, or other institutional body to which data requests may be sent.

2b) If there are no restrictions, please upload the minimal anonymized data set necessary to replicate your study findings as either Supporting Information files or to a stable, public repository and provide us with the relevant URLs, DOIs, or accession numbers. Please see http://www.bmj.com/content/340/bmj.c181.long for guidelines on how to de-identify and prepare clinical data for publication. For a list of acceptable repositories, please see http://journals.plos.org/plosone/s/data-availability#loc-recommended-repositories.

Include this sentence at the end of your statement: The funders had no role in study design, data collection and analysis, decision to publish, or preparation of the manuscript.

3a)           Please clarify the sources of funding (financial or material support) for your study. List the grants or organizations that supported your study, including funding received from your institution.

3b)           State what role the funders took in the study. If the funders had no role in your study, please state: “The funders had no role in study design, data collection and analysis, decision to publish, or preparation of the manuscript.”

3c)           If any authors received a salary from any of your funders, please state which authors and which funders.

3d)           If you did not receive any funding for this study, please state: “The authors received no specific funding for this work.”

Reviewers' comments:

Reviewer's Responses to Questions

**Comments to the Author**

1. Is the manuscript technically sound, and do the data support the conclusions?

Reviewer #1: Yes

Reviewer #2: Yes

2. Has the statistical analysis been performed appropriately and rigorously? 

Reviewer #1: Yes

Reviewer #2: Yes

3. Have the authors made all data underlying the findings in their manuscript fully available?

Reviewer #1: No

Reviewer #2: Yes

4. Is the manuscript presented in an intelligible fashion and written in standard English?

Reviewer #1: Yes

Reviewer #2: Yes

5. Review Comments to the Author

Reviewer #1: The article presents original data from a large sample and results are in acordance with previous studies.

1. More detailed information should be made available about the scale used to assess mental health symptoms, as it appears from the manuscript that this was a 90 itens questionnaire. HSCL has also a 10 and a 25 questions version, correct? This is particularly important as 3 out of the six questions used to define the outcome variable could also be interpreted as signs of anxiety (itens 2, 5 and 6), not being specific to depression. Please clarify.

2. If more information was gathered on mental health symptoms, it would be interesting to see the association between attentional problems and gaming, which is frequently discussed in the literature.

3. A few small corrections in the text:

- In Table 1, response alternatives to social media use and gaming have the "> sign" inverted (>3 hours)

- In Table 1, parents higher education, the word parents is misspelled

- In Table 1, the variable definition does not match the response alternatives in social media use and gaming (times per day vs hours per day).

- Line 105-106: the word derived appears twice

- Line 222: reference style

4. In the logistic regression (table 3), the ORs of depressive symptoms for both gender and economic status are much higher than those for social media use or gaming. I would recommend, in the discussion, commenting/highlighting the importance of <<multiple factors="">> that possibly mediate, moderate or confound the association between screen media use and mental health issues.

5. Why did the authors predict an interaction effect between gaming and social media use? Have you run separate models for boys and girls? Based on the literature, I would expect that the strength of the relationship between electronic media use and symptoms of depression would be affected by socioeconomic status

6. Exchanging text, audio or video messages with friends through WhatsApp, Messenger or similar apps was included as social media use? (sorry, I don't know if these are used in Norway). Did the question presented to the adolescents mention plataforms other than Facebook and Instagram?</multiple>

Reviewer #2: Overall, this is a very clear and concise study on the associations of electronic media use and depression in adolescents. The results imply that more time using social media or video games is associated with a greater depression risk. The paper is really well-written, easy to follow, and addresses simple but important research questions - congratulations on such a nice manuscript to read!

My main reservations about this paper are the lack of adjustment for several key confounding variables (highlighted below) and the unnecessary dichotomisation of the exposure and outcome variables. I explain these points below, along with some other more minor additions:

Introduction

- Page 3 final paragraph, it's worth mentioning a meta-analysis by Liu et al 2016 on screen time and depressive symptoms in young people found no association between video gaming and depression (but did with other forms of screen time) in a sub group analysis - these seems to contrast with your Zink reference

- Liu M, Wu L, Yao S. Dose-response association of screen time-based sedentary behaviour in children and adolescents and depression: a meta-analysis of observational studies. Br J Sports Med [Internet]. 2016 Oct 1 [cited 2019 Sep 17];50(20):1252–8. Available from: http://www.ncbi.nlm.nih.gov/pubmed/26552416

- There was also a recent longitudinal study that showed protective associations between video game use and depressive symptoms, which did find gender as a possible moderator (as I believe did Liu et al) in contrast to lines 71-72:

- Kandola A, Owen N, Dunstan DW, Hallgren M. Prospective relationships of adolescents’ screen-based sedentary behaviour with depressive symptoms: the Millennium Cohort Study. Psychol Med [Internet]. 2021 Feb 19 [cited 2021 Mar 1];1–9. Available from: http://www.ncbi.nlm.nih.gov/pubmed/33602369

- Another recent (cross sectional) study which used objective measures of video gaming found protective associations with wellbeing, which seems relevant here:

- Johannes, N., Vuorre, M., & Przybylski, A. K. (2020). Video game play is positively correlated with well-being. Royal Society Open Science.

Methods:

- Do you have data on response rates and/or attrition/missing data for the survey? It would be helpful for assessing the extend of selection bias

- Table 1 could probably go in the supplementary materials

- Electronic media use - why dichotomise these variables? Could the authors provide some clear justification for this. I would advise against it here, you lose a lot of information this way and these individual categories represent different usage patterns that warrant investigating e.g., under an hour or 1-2 hours

- Depressive symptoms - again I'd argue against dichotomising here. Depressive symptoms exist on a continuum in reality not categorically. This is particularly relevant here as you're using a depression scale that isn't widely used as a screening tool for depression. You also sacrifice statistical power e.g., see Royston P, Altman DG, Sauerbrei W. Dichotomizing continuous predictors in multiple regression: a bad idea. Stat Med [Internet]. 2006 Jan 15 [cited 2019 May 29];25(1):127–41. Available from: http://doi.wiley.com/10.1002/sim.2331

- Can the authors state how these confounding variables were selected? There are many potential confounding variables that aren't included here, e.g., household income/deprivation, time in physical activity/sedentary behaviour, alcohol use, parental mental health or another marker of genetic mental health risk, overall physical health or disability

Results:

- Table 2:  There is quite a lot going on in this table. I suggest shortening the variable names for each characteristics to just 'social media', 'video game' etc. And only comparing across gender or depressive symptoms rather than both

- Table 3:  I am unclear what is being presented here. It looks like these were all separate univariate models in the way they are presented here. Presumably this is the main analysis where you ran social media and gaming as exposure variables (in separate models? or mutually adjusted?) with depression as the outcome, adjusted for the confounders? If so, I suggest first including the crude (unadjusted) estimates then the fully adjusted estimates - only showing the ORs for the exposures (i.e., gaming and social media). The ORs for the confounding variables can be misleading for some readers as they are commonly misinterpreted, see this paper for a clear explanation of this as the 'Table 2 fallacy':

- Westreich, D., & Greenland, S. (2013). The table 2 fallacy: presenting and interpreting confounder and modifier coefficients. American journal of epidemiology, 177(4), 292-298.

- I'd also include a sentence clarifying there was no interactions with gender and friends with the appropriate statistics for indicating this (rather than indicating only in the methods)

Discussion

- The first couple of sentence can be moved down or removed, you just need a clear sentence clarifying the aim of this study followed swiftly by the main findings

- Lines 227-228, again worth mentioning this contrasts with the studies I mentioned to include in the discussion finding protective associations of video gaming on mental health - I also think this merits some discussion, why do the authors think these results contrast?

- lines 253 - include any sensitivity analysis in the methods and results, not in the discussion alone

- Strengths and limitations needs a bit of expansion, this is an important part of the paper for guiding future research. I'd mention the sample size as a key strength. Weaknesses to consider include:

- the self-report exposure measures that are subject to recall and social desirability biases - worth noting the Johannes paper mentioned above used objective measures and found the opposite association.

- exposure data is only focusing on time, whereas contextual factors are likely to be equally as important to its relationship with depression e.g., what type of video games were participants playing? what were they using social media to do? Interact with friends or scroll through newsfeeds aimlessly?

- no data on television - which is the most studied domain of screen time in adults

- The depressive symptoms are measured using a not particularly well validated tool, which could introduce additional measurement error

- There are also several confounding variables missing that I mention above as limitations

- Conclusion is also a little short, id include a sentence stating the implications/future directions for research

6. PLOS authors have the option to publish the peer review history of their article (what does this mean?). If published, this will include your full peer review and any attached files.

Reviewer #1: **Yes: **Renata Kieling

Reviewer #2: **Yes: **Aaron Kandola

---

## [Author Response · Author response to Decision Letter 0]

2 Jun 2021

Reviewer #1: The article presents original data from a large sample and results are in accordance with previous studies.

1. More detailed information should be made available about the scale used to assess mental health symptoms, as it appears from the manuscript that this was a 90 items questionnaire. HSCL has also a 10 and a 25 questions version, correct? This is particularly important as 3 out of the six questions used to define the outcome variable could also be interpreted as signs of anxiety (itens 2, 5 and 6), not being specific to depression. Please clarify.

We have added more text in the manuscript to clarify (page 6 under Measures).

Some of the items measuring depressive symptoms in the HSCL-10 are similar to the depressive symptom scale, but the wording on some of the items are different. We have chosen not to use items measuring anxiety in the current study. A reason for that is that anxiety items from the HSCL-10 is not in the main module in Ungdata, only in one of the optional modules which some schools have chosen not to include.

2. If more information was gathered on mental health symptoms, it would be interesting to see the association between attentional problems and gaming, which is frequently discussed in the literature.

We agree that this could be interesting to examine. However, according to the aim of our study we chose to focus on symptoms of depression in the current study and use the validated depressive symptoms scale. Please see the explanation above.

1. A few small corrections in the text:

- In Table 1, response alternatives to social media use and gaming have the "> sign" inverted (>3 hours)

- In Table 1, parents higher education, the word parents is misspelled

- In Table 1, the variable definition does not match the response alternatives in social media use and gaming (times per day vs hours per day).

- Line 105-106: the word derived appears twice

- Line 222: reference style

Thank you, the changes are made.

4. In the logistic regression (table 3), the ORs of depressive symptoms for both gender and economic status are much higher than those for social media use or gaming. I would recommend, in the discussion, commenting/highlighting the importance of <> that possibly mediate, moderate or confound the association between screen media use and mental health issues.

Table 3 is revised after comments from the other reviewer. In the method section (page 9), the moderator analysis in the present study is described and further discussed on page 14, line 253-264.

5. Why did the authors predict an interaction effect between gaming and social media use? Have you run separate models for boys and girls? Based on the literature, I would expect that the strength of the relationship between electronic media use and symptoms of depression would be affected by socioeconomic status

Zink et. al. (2020) indicated in a recent systematic review of moderating variables that screen-type influences the strength of the association between screen based sedentary behaviour and symptoms of depression and anxiety in adolescence. Therefore, we did interaction analysis to examine if the strength of the association between social media use and depressive symptoms was affected by gaming. We performed separate analysis for boys and girls, we did not find anything new, and the results for boys and girls are consistent with the results presented in this study. 

Following the reviewer’s suggestion, we have conducted new interaction analysis to examine if the strength of the association between electronic media use and symptoms of depression would be affected by socioeconomic status. The incremental changes in log-likelihood between the main effect models and models including interactions were not significant implying that the fit was not improved when applying interaction models. 

6. Exchanging text, audio or video messages with friends through WhatsApp, Messenger or similar apps was included as social media use? (sorry, I don't know if these are used in Norway). Did the question presented to the adolescents mention plataforms other than Facebook and Instagram?

The national Ungdata used in the present study mentioned only Facebook, Instagram and etc.

Reviewer #2: Overall, this is a very clear and concise study on the associations of electronic media use and depression in adolescents. The results imply that more time using social media or video games is associated with a greater depression risk. The paper is really well-written, easy to follow, and addresses simple but important research questions - congratulations on such a nice manuscript to read!

My main reservations about this paper are the lack of adjustment for several key confounding variables (highlighted below) and the unnecessary dichotomisation of the exposure and outcome variables. I explain these points below, along with some other more minor additions:

Introduction

- Page 3 final paragraph, it's worth mentioning a meta-analysis by Liu et al 2016 on screen time and depressive symptoms in young people found no association between video gaming and depression (but did with other forms of screen time) in a sub group analysis - these seems to contrast with your Zink reference

- Liu M, Wu L, Yao S. Dose-response association of screen time-based sedentary behaviour in children and adolescents and depression: a meta-analysis of observational studies. Br J Sports Med [Internet]. 2016 Oct 1 [cited 2019 Sep 17];50(20):1252–8. Available from: http://www.ncbi.nlm.nih.gov/pubmed/26552416

- There was also a recent longitudinal study that showed protective associations between video game use and depressive symptoms, which did find gender as a possible moderator (as I believe did Liu et al) in contrast to lines 71-72:

- Kandola A, Owen N, Dunstan DW, Hallgren M. Prospective relationships of adolescents’ screen-based sedentary behaviour with depressive symptoms: the Millennium Cohort Study. Psychol Med [Internet]. 2021 Feb 19 [cited 2021 Mar 1];1–9. Available from: http://www.ncbi.nlm.nih.gov/pubmed/33602369

- Another recent (cross sectional) study which used objective measures of video gaming found protective associations with wellbeing, which seems relevant here:

- Johannes, N., Vuorre, M., & Przybylski, A. K. (2020). Video game play is positively correlated with well-being. Royal Society Open Science.

Thank you for addressing this, new references are added, and changes are made in the manuscript (page 3 and 4).

Methods:

- Do you have data on response rates and/or attrition/missing data for the survey? It would be helpful for assessing the extend of selection bias

We agree and have added the participation rate for secondary school (85%). Additionally, in table 2 you can see how many adolescents who have answered on the questions used in this study.

- Table 1 could probably go in the supplementary materials

We have chosen to keep the table, because we think it clarify the text. 

- Electronic media use - why dichotomise these variables? Could the authors provide some clear justification for this. I would advise against it here, you lose a lot of information this way and these individual categories represent different usage patterns that warrant investigating e.g., under an hour or 1-2 hours

We understand your advice. Adolescents nowadays spend more time on online gaming and social media than previous generations, and recent studies have used the same cut off value <3h per day (Boshi et al 2020 & Kleppang et al 2021). 

Nonetheless, to address this comment, we have taken the following actions. 

1) In the Analysis section, we added information regarding testing robustness of our results (see page 10).

2) At the end of the Results section, we added information regarding results of our robustness test (see page 13).

3) In the section of Discussion, we discussed the results from the robustness test (see page 15). 

- Depressive symptoms - again I'd argue against dichotomising here. Depressive symptoms exist on a continuum in reality not categorically. This is particularly relevant here as you're using a depression scale that isn't widely used as a screening tool for depression. You also sacrifice statistical power e.g., see Royston P, Altman DG, Sauerbrei W. Dichotomizing continuous predictors in multiple regression: a bad idea. Stat Med [Internet]. 2006 Jan 15 [cited 2019 May 29];25(1):127–41. Available from: http://doi.wiley.com/10.1002/sim.2331

The scale has been used in The Municipal Youth Surveys (Ungdata) since 2010. A psychometric analysis of the scale has been done using Rasch measurement theory, and the dichotomising of the depression scale has been used in previous studies. We have added text and ref. to clarify in the method section on page 6.

- Can the authors state how these confounding variables were selected? There are many potential confounding variables that aren't included here, e.g., household income/deprivation, time in physical activity/sedentary behaviour, alcohol use, parental mental health or another marker of genetic mental health risk, overall physical health or disability

Existing studies have shown that SES and smoking are associated with depression. Additionally, friendship have been reported to be inversely associated with depression. Hence, we used parent’s higher education, family economy, gender and having friends as confounding variables. 

Results:

- Table 2: There is quite a lot going on in this table. I suggest shortening the variable names for each characteristics to just 'social media', 'video game' etc. And only comparing across gender or depressive symptoms rather than both

We agree and have shortened the variable names. 

- Table 3: I am unclear what is being presented here. It looks like these were all separate univariate models in the way they are presented here. Presumably this is the main analysis where you ran social media and gaming as exposure variables (in separate models? or mutually adjusted?) with depression as the outcome, adjusted for the confounders? If so, I suggest first including the crude (unadjusted) estimates then the fully adjusted estimates - only showing the ORs for the exposures (i.e., gaming and social media). The ORs for the confounding variables can be misleading for some readers as they are commonly misinterpreted, see this paper for a clear explanation of this as the 'Table 2 fallacy':

- Westreich, D., & Greenland, S. (2013). The table 2 fallacy: presenting and interpreting confounder and modifier coefficients. American journal of epidemiology, 177(4), 292-298.

We agree and have made the changes in the table.

- I'd also include a sentence clarifying there was no interactions with gender and friends with the appropriate statistics for indicating this (rather than indicating only in the methods)

We agree and have added this in the results on page 13.

Discussion

- The first couple of sentence can be moved down or removed, you just need a clear sentence clarifying the aim of this study followed swiftly by the main findings

We agree and have moved down the sentences to the introduction part (page 13, line 231-235).

- Lines 227-228, again worth mentioning this contrasts with the studies I mentioned to include in the discussion finding protective associations of video gaming on mental health - I also think this merits some discussion, why do the authors think these results contrast?

We agree and have added the articles in the discussion.

- lines 253 - include any sensitivity analysis in the methods and results, not in the discussion alone

We agree and have added this both under method and results section.

- Strengths and limitations needs a bit of expansion, this is an important part of the paper for guiding future research. I'd mention the sample size as a key strength. Weaknesses to consider include:

We agree and have added the following: The present study’s major strengths were the large sample size combined with a high response rate.

- the self-report exposure measures that are subject to recall and social desirability biases - worth noting the Johannes paper mentioned above used objective measures and found the opposite association.

We agree and have added this as a limitation

- exposure data is only focusing on time, whereas contextual factors are likely to be equally as important to its relationship with depression e.g., what type of video games were participants playing? what were they using social media to do? Interact with friends or scroll through newsfeeds aimlessly?

We agree and have added this as a limitation.

- no data on television - which is the most studied domain of screen time in adults

Since the present study have examined adolescents, we chose to focus on electronic media use.

- The depressive symptoms are measured using a not particularly well validated tool, which could introduce additional measurement error

The scale has been validated in the present study and in previous studies (see page 7, line 120-125) 

We have added that the scale has been validated using Rasch Measurement Theory (see page 15).

- There are also several confounding variables missing that I mention above as limitations

We have added this as a limitation

- Conclusion is also a little short, id include a sentence stating the implications/future directions for research

We agree and have added sentence to clarify (see page 15)

---

## [Decision Letter · Decision Letter 1]

23 Jun 2021

Electronic media use and symptoms of depression among adolescents in Norway

PONE-D-20-34067R1

Dear Dr. Kleppang,

We’re pleased to inform you that your manuscript has been judged scientifically suitable for publication and will be formally accepted for publication once it meets all outstanding technical requirements.

Kind regards,

Therese van Amelsvoort

Academic Editor

PLOS ONE

Additional Editor Comments (optional):

Reviewers' comments:

Reviewer's Responses to Questions

**Comments to the Author**

1. If the authors have adequately addressed your comments raised in a previous round of review and you feel that this manuscript is now acceptable for publication, you may indicate that here to bypass the “Comments to the Author” section, enter your conflict of interest statement in the “Confidential to Editor” section, and submit your "Accept" recommendation.

Reviewer #1: All comments have been addressed

2. Is the manuscript technically sound, and do the data support the conclusions?

Reviewer #1: Yes

3. Has the statistical analysis been performed appropriately and rigorously? 

Reviewer #1: Yes

4. Have the authors made all data underlying the findings in their manuscript fully available?

Reviewer #1: Yes

5. Is the manuscript presented in an intelligible fashion and written in standard English?

Reviewer #1: Yes

6. Review Comments to the Author

Reviewer #1: All relevant comments have been properly addressed and I think the manuscript is ready for publication.

7. PLOS authors have the option to publish the peer review history of their article (what does this mean?). If published, this will include your full peer review and any attached files.

Reviewer #1: No

---

## [Editor Report · Acceptance letter]

28 Jun 2021

PONE-D-20-34067R1 

Electronic media use and symptoms of depression among adolescents in Norway 

Dear Dr. Kleppang:

I'm pleased to inform you that your manuscript has been deemed suitable for publication in PLOS ONE. Congratulations! Your manuscript is now with our production department. 

Kind regards, 

on behalf of

Prof. Therese van Amelsvoort 

Academic Editor

PLOS ONE